# Good Semi-supervised Learning
# That Requires a Bad GAN

**Zihang Dai**[*], **Zhilin Yang**[*], **Fan Yang, William W. Cohen, Ruslan Salakhutdinov**
School of Computer Science
Carnegie Melon University
`dzihang,zhiliny,fanyang1,wcohen,rsalakhu@cs.cmu.edu`

## Abstract

Semi-supervised learning methods based on generative adversarial networks (GANs) obtained strong empirical results, but it is not clear 1) how the discriminator benefits from joint training with a generator, and 2) why good semi-supervised classification performance and a good generator cannot be obtained at the same time. Theoretically we show that given the discriminator objective, good semi-supervised learning indeed requires a bad generator, and propose the definition of a preferred generator. Empirically, we derive a novel formulation based on our analysis that substantially improves over feature matching GANs, obtaining state-of-the-art results on multiple benchmark datasets[2].

## 1   Introduction

Deep neural networks are usually trained on a large amount of labeled data, and it has been a challenge to apply deep models to datasets with limited labels. Semi-supervised learning (SSL) aims to leverage the large amount of unlabeled data to boost the model performance, particularly focusing on the setting where the amount of available labeled data is limited. Traditional graph-based methods [2, 26] were extended to deep neural networks [22, 23, 8], which involves applying convolutional neural networks [10] and feature learning techniques to graphs so that the underlying manifold structure can be exploited. [15] employs a Ladder network to minimize the layerwise reconstruction loss in addition to the standard classification loss. Variational auto-encoders have also been used for semi-supervised learning [7, 12] by maximizing the variational lower bound of the unlabeled data log-likelihood.

Recently, generative adversarial networks (GANs) [6] were demonstrated to be able to generate visually realistic images. GANs set up an adversarial game between a discriminator and a generator. The goal of the discriminator is to tell whether a sample is drawn from true data or generated by the generator, while the generator is optimized to generate samples that are not distinguishable by the discriminator. Feature matching (FM) GANs [16] apply GANs to semi-supervised learning on $K$-class classification. The objective of the generator is to match the first-order feature statistics between the generator distribution and the true distribution. Instead of binary classification, the discriminator employs a $(K + 1)$-class objective, where true samples are classified into the first $K$ classes and generated samples are classified into the $(K + 1)$-th class. This $(K + 1)$-class discriminator objective leads to strong empirical results, and was later widely used to evaluate the effectiveness of generative models [5, 21].

Though empirically feature matching improves semi-supervised classification performance, the following questions still remain open. First, it is not clear why the formulation of the discriminator

---

[*]Equal contribution. Ordering determined by dice rolling.
[2]Code is available at `https://github.com/kimiyoung/ssl_bad_gan`.

can improve the performance when combined with a generator. Second, it seems that good semi-supervised learning and a good generator cannot be obtained at the same time. For example, [16] observed that mini-batch discrimination generates better images than feature matching, but feature matching obtains a much better semi-supervised learning performance. The same phenomenon was also observed in [21], where the model generated better images but failed to improve the performance on semi-supervised learning.

In this work, we take a step towards addressing these questions. First, we show that given the current $(K + 1)$-class discriminator formulation of GAN-based SSL, good semi-supervised learning requires a "bad" generator. Here by *bad* we mean the generator distribution should not match the true data distribution. Then, we give the definition of a preferred generator, which is to generate complement samples in the feature space. Theoretically, under mild assumptions, we show that a properly optimized discriminator obtains correct decision boundaries in high-density areas in the feature space if the generator is a *complement generator*.

Based on our theoretical insights, we analyze why feature matching works on 2-dimensional toy datasets. It turns out that our practical observations align well with our theory. However, we also find that the feature matching objective has several drawbacks. Therefore, we develop a novel formulation of the discriminator and generator objectives to address these drawbacks. In our approach, the generator minimizes the KL divergence between the generator distribution and a target distribution that assigns high densities for data points with low densities in the true distribution, which corresponds to the idea of a complement generator. Furthermore, to enforce our assumptions in the theoretical analysis, we add the conditional entropy term to the discriminator objective.

Empirically, our approach substantially improves over vanilla feature matching GANs, and obtains new state-of-the-art results on MNIST, SVHN, and CIFAR-10 when all methods are compared under the same discriminator architecture. Our results on MNIST and SVHN also represent state-of-the-art amongst all single-model results.

## 2    Related Work

Besides the adversarial feature matching approach [16], several previous works have incorporated the idea of adversarial training in semi-supervised learning. Notably, [19] proposes categorical generative adversarial networks (CatGAN), which substitutes the binary discriminator in standard GAN with a multi-class classifier, and trains both the generator and the discriminator using information theoretical criteria on unlabeled data. From the perspective of regularization, [14, 13] propose virtual adversarial training (VAT), which effectively smooths the output distribution of the classifier by seeking virtually adversarial samples. It is worth noting that VAT bears a similar merit to our approach, which is to learn from auxiliary non-realistic samples rather than realistic data samples. Despite the similarity, the principles of VAT and our approach are orthogonal, where VAT aims to enforce a smooth function while we aim to leverage a generator to better detect the low-density boundaries. Different from aforementioned approaches, [24] proposes to train conditional generators with adversarial training to obtain complete sample pairs, which can be directly used as additional training cases. Recently, Triple GAN [11] also employs the idea of conditional generator, but uses adversarial cost to match the two model-defined factorizations of the joint distribution with the one defined by paired data.

Apart from adversarial training, there has been other efforts in semi-supervised learning using deep generative models recently. As an early work, [7] adapts the original Variational Auto-Encoder (VAE) to a semi-supervised learning setting by treating the classification label as an additional latent variable in the directed generative model. [12] adds auxiliary variables to the deep VAE structure to make variational distribution more expressive. With the boosted model expressiveness, auxiliary deep generative models (ADGM) improve the semi-supervised learning performance upon the semi-supervised VAE. Different from the explicit usage of deep generative models, the Ladder networks [15] take advantage of the local (layerwise) denoising auto-encoding criterion, and create a more informative unsupervised signal through lateral connection.

## 3    Theoretical Analysis

Given a labeled set $\mathcal{L} = \{(x, y)\}$, let $\{1, 2, \cdots, K\}$ be the label space for classification. Let $D$ and $G$ denote the discriminator and generator, and $P_D$ and $p_G$ denote the corresponding distributions.

Consider the discriminator objective function of GAN-based semi-supervised learning [16]:

$$\max_D \mathbb{E}_{x,y \sim \mathcal{L}} \log P_D(y|x, y \leq K) + \mathbb{E}_{x \sim p} \log P_D(y \leq K|x) + \mathbb{E}_{x \sim p_G} \log P_D(K+1|x), \quad (1)$$

where $p$ is the true data distribution. The probability distribution $P_D$ is over $K+1$ classes where the first $K$ classes are true classes and the $(K+1)$-th class is the fake class. The objective function consists of three terms. The first term is to maximize the log conditional probability for labeled data, which is the standard cost as in supervised learning setting. The second term is to maximize the log probability of the first $K$ classes for unlabeled data. The third term is to maximize the log probability of the $(K+1)$-th class for generated data. Note that the above objective function bears a similar merit to the original GAN formulation if we treat $P(K+1|x)$ to be the probability of fake samples, while the only difference is that we split the probability of true samples into $K$ sub-classes.

Let $f(x)$ be a nonlinear vector-valued function, and $w_k$ be the weight vector for class $k$. As a standard setting in previous work [16, 5], the discriminator $D$ is defined as $P_D(k|x) = \frac{\exp(w_k^\top f(x))}{\sum_{k'=1}^{K+1} \exp(w_{k'}^\top f(x))}$. Since this is a form of over-parameterization, $w_{K+1}$ is fixed as a zero vector [16]. We next discuss the choices of different possible $G$'s.

## 3.1 Perfect Generator

Here, by perfect generator we mean that the generator distribution $p_G$ exactly matches the true data distribution $p$, i.e., $p_G = p$. We now show that when the generator is perfect, it does not improve the generalization over the supervised learning setting.

**Proposition 1.** *If $p_G = p$, and $D$ has infinite capacity, then for any optimal solution $D = (w, f)$ of the following supervised objective,*

$$\max_D \mathbb{E}_{x,y \sim \mathcal{L}} \log P_D(y|x, y \leq K), \quad (2)$$

*there exists $D^* = (w^*, f^*)$ such that $D^*$ maximizes Eq. (1) and that for all $x$, $P_D(y|x, y \leq K) = P_{D^*}(y|x, y \leq K)$.*

The proof is provided in the supplementary material. Proposition 1 states that for any optimal solution $D$ of the supervised objective, there exists an optimal solution $D^*$ of the $(K+1)$-class objective such that $D$ and $D^*$ share the same generalization error. In other words, using the $(K+1)$-class objective does not prevent the model from experiencing any arbitrarily high generalization error that it could suffer from under the supervised objective. Moreover, since all the optimal solutions are equivalent w.r.t. the $(K+1)$-class objective, it is the optimization algorithm that really decides which specific solution the model will reach, and thus what generalization performance it will achieve. This implies that when the generator is perfect, the $(K+1)$-class objective by itself is not able to improve the generalization performance. In fact, in many applications, an almost infinite amount of unlabeled data is available, so learning a perfect generator for purely sampling purposes should not be useful. In this case, our theory suggests that not only the generator does not help, but also unlabeled data is not effectively utilized when the generator is perfect.

## 3.2 Complement Generator

The function $f$ maps data points in the input space to the feature space. Let $p_k(f)$ be the density of the data points of class $k$ in the feature space. Given a threshold $\epsilon_k$, let $F_k$ be a subset of the data support where $p_k(f) > \epsilon_k$, i.e., $F_k = \{f : p_k(f) > \epsilon_k\}$. We assume that given $\{\epsilon_k\}_{k=1}^{K}$, the $F_k$'s are disjoint with a margin. More formally, for any $f_j \in F_j$, $f_k \in F_k$, and $j \neq k$, we assume that there exists a real number $0 < \alpha < 1$ such that $\alpha f_j + (1-\alpha)f_k \notin F_j \cup F_k$. As long as the probability densities of different classes do not share any mode, i.e., $\forall i \neq j, \mathrm{argmax}_f p_i(f) \cap \mathrm{argmax}_f p_j(f) = \emptyset$, this assumption can always be satisfied by tuning the thresholds $\epsilon_k$'s. With the assumption held, we will show that the model performance would be better if the thresholds could be set to smaller values (ideally zero). We also assume that each $F_k$ contains at least one labeled data point.

Suppose $\cup_{k=1}^{K} F_k$ is bounded by a convex set $\mathcal{B}$. If the support $F_G$ of a generator $G$ in the feature space is a relative complement set in $\mathcal{B}$, i.e., $F_G = \mathcal{B} - \cup_{k=1}^{K} F_k$, we call $G$ a complement generator. The reason why we utilize a bounded $\mathcal{B}$ to define the complement is presented in the supplementary

material. Note that the definition of complement generator implies that $G$ is a function of $f$. By treating $G$ as function of $f$, theoretically $D$ can optimize the original objective function in Eq. (1).

Now we present the assumption on the convergence conditions of the discriminator. Let $\mathcal{U}$ and $\mathcal{G}$ be the sets of unlabeled data and generated data.

**Assumption 1.** *Convergence conditions. When $D$ converges on a finite training set $\{\mathcal{L}, \mathcal{U}, \mathcal{G}\}$, $D$ learns a (strongly) correct decision boundary for all training data points. More specifically, (1) for any $(x, y) \in \mathcal{L}$, we have $w_y^\top f(x) > w_k^\top f(x)$ for any other class $k \neq y$; (2) for any $x \in \mathcal{G}$, we have $0 > \max_{k=1}^K w_k^\top f(x)$; (3) for any $x \in \mathcal{U}$, we have $\max_{k=1}^K w_k^\top f(x) > 0$.*

In Assumption 1, conditions (1) and (2) assume classification correctness on labeled data and true-fake correctness on generated data respectively, which is directly induced by the objective function. Likewise, it is also reasonable to assume true-fake correctness on unlabeled data, i.e., $\log \sum_k \exp w_k^\top f(x) > 0$ for $x \in \mathcal{U}$. However, condition (3) goes beyond this and assumes $\max_k w_k^\top f(x) > 0$. We discuss this issue in detail in the supplementary material and argue that these assumptions are reasonable. Moreover, in Section 5, our approach addresses this issue explicitly by adding a conditional entropy term to the discriminator objective to enforce condition (3).

**Lemma 1.** *Suppose for all $k$, the L2-norms of weights $w_k$ are bounded by $\|w_k\|_2 \leq C$. Suppose that there exists $\epsilon > 0$ such that for any $f_G \in F_G$, there exists $f_G' \in \mathcal{G}$ such that $\|f_G - f_G'\|_2 \leq \epsilon$. With the conditions in Assumption 1, for all $k \leq K$, we have $w_k^\top f_G < C\epsilon$.*

**Corollary 1.** *When unlimited generated data samples are available, with the conditions in Lemma 1, we have $\lim_{|\mathcal{G}| \to \infty} w_k^\top f_G \leq 0$.*

See the supplementary material for the proof.

**Proposition 2.** *Given the conditions in Corollary 1, for all class $k \leq K$, for all feature space points $f_k \in F_k$, we have $w_k^\top f_k > w_j^\top f_k$ for any $j \neq k$.*

*Proof.* Without loss of generality, suppose $j = \arg\max_{j \neq k} w_j^\top f_k$. Now we prove it by contradiction. Suppose $w_k^\top f_k \leq w_j^\top f_k$. Since $F_k$'s are disjoint with a margin, $\mathcal{B}$ is a convex set and $F_G = \mathcal{B} - \cup_k F_k$, there exists $0 < \alpha < 1$ such that $f_G = \alpha f_k + (1 - \alpha)f_j$ with $f_G \in F_G$ and $f_j$ being the feature of a labeled data point in $F_j$. By Corollary 1, it follows that $w_j^\top f_G \leq 0$. Thus, $w_j^\top f_G = \alpha w_j^\top f_k + (1 - \alpha)w_j^\top f_j \leq 0$. By Assumption 1, $w_j^\top f_k > 0$ and $w_j^\top f_j > 0$, leading to contradiction. It follows that $w_k^\top f_k > w_j^\top f_k$ for any $j \neq k$. $\qquad\square$

Proposition 2 guarantees that when $G$ is a complement generator, under mild assumptions, a near-optimal $D$ learns correct decision boundaries in each high-density subset $F_k$ (defined by $\epsilon_k$) of the data support in the feature space. Intuitively, the generator generates complement samples so the logits of the true classes are forced to be low in the complement. As a result, the discriminator obtains class boundaries in low-density areas. This builds a connection between our approach with manifold-based methods [2, 26] which also leverage the low-density boundary assumption.

With our theoretical analysis, we can now answer the questions raised in Section 1. First, the $(K + 1)$-class formulation is effective because the generated complement samples encourage the discriminator to place the class boundaries in low-density areas (Proposition 2). Second, good semi-supervised learning indeed requires a bad generator because a perfect generator is not able to improve the generalization performance (Proposition 1).

## 4 Case Study on Synthetic Data

In the previous section, we have established the fact a complement generator, instead of a perfect generator, is what makes a good semi-supervised learning algorithm. Now, to get a more intuitive understanding, we conduct a case study based on two 2D synthetic datasets, where we can easily verify our theoretical analysis by visualizing the model behaviors. In addition, by analyzing how feature matching (FM) [16] works in 2D space, we identify some potential problems of it, which motivates our approach to be introduced in the next section. Specifically, two synthetic datasets are four spins and two circles, as shown in Fig. 1.

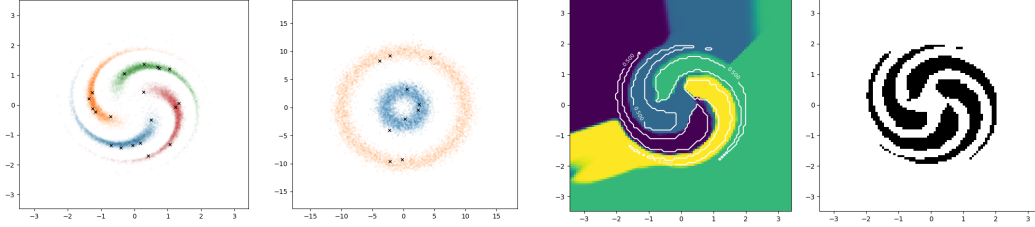

Figure 1: Labeled and unlabeled data are denoted by cross and point respectively, and different colors indicate classes.

Figure 2: Left: Classification decision boundary, where the white line indicates true-fake boundary; Right: True-Fake decision boundary

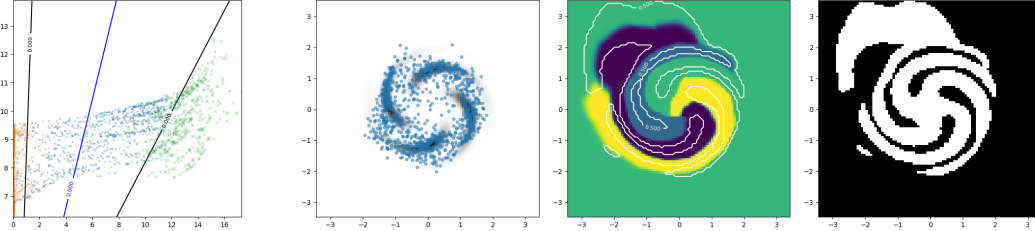

Figure 3: Feature space at convergence

Figure 4: Left: Blue points are generated data, and the black shadow indicates unlabeled data. Middle and right can be interpreted as above.

**Soundness of complement generator** Firstly, to verify that the complement generator is a preferred choice, we construct the complement generator by uniformly sampling from the a bounded 2D box that contains all unlabeled data, and removing those on the manifold. Based on the complement generator, the result on four spins is visualized in Fig. 2. As expected, both the classification and true-fake decision boundaries are almost perfect. More importantly, the classification decision boundary always lies in the fake data area (left panel), which well matches our theoretical analysis.

**Visualization of feature space** Next, to verify our analysis about the feature space, we choose the feature dimension to be 2, apply the FM to the simpler dataset of two circles, and visualize the feature space in Fig. 3. As we can see, most of the generated features (blue points) resides in between the features of two classes (green and orange crosses), although there exists some overlap. As a result, the discriminator can almost perfectly distinguish between true and generated samples as indicated by the black decision boundary, satisfying the our required Assumption 1. Meanwhile, the model obtains a perfect classification boundary (blue line) as our analysis suggests.

**Pros and cons of feature matching** Finally, to further understand the strength and weakness of FM, we analyze the solution FM reaches on four spins shown in Fig. 4. From the left panel, we can see many of the generated samples actually fall into the data manifold, while the rest scatters around in the nearby surroundings of data manifold. It suggests that by matching the first-order moment by SGD, FM is performing some kind of distribution matching, though in a rather *weak* manner. Loosely speaking, FM has the effect of generating samples close to the manifold. But due to its weak power in distribution matching, FM will inevitably generate samples outside of the manifold, especially when the data complexity increases. Consequently, the generator density $p_G$ is usually lower than the true data density $p$ within the manifold and higher outside. Hence, an optimal discriminator $P_{D^*}(K + 1 \mid x) = p(x)/(p(x) + p_G(x))$ could still distinguish between true and generated samples in many cases. However, there are two types of mistakes the discriminator can still make

1. Higher density mistake inside manifold: Since the FM generator still assigns a significant amount of probability mass inside the support, wherever $p_G > p > 0$, an optimal discriminator will incorrectly predict samples in that region as "fake". Actually, this problem has already shown up when we examine the feature space (Fig. 3).

2. Collapsing with missing coverage outside manifold: As the feature matching objective for the generator only requires matching the first-order statistics, there exists many trivial solutions the generator can end up with. For example, it can simply collapse to mean of unlabeled features, or a few surrounding modes as along as the feature mean matches. Actually, we do see such

collapsing phenomenon in high-dimensional experiments when FM is used (see Fig. 5a and Fig. 5c) As a result, a collapsed generator will fail to cover some gap areas between manifolds. Since the discriminator is only well-defined on the union of the data supports of $p$ and $p_G$, the prediction result in such missing area is under-determined and fully relies on the smoothness of the parametric model. In this case, significant mistakes can also occur.

## 5 Approach

As discussed in previous sections, feature matching GANs suffer from the following drawbacks: 1) the first-order moment matching objective does not prevent the generator from collapsing (missing coverage); 2) feature matching can generate high-density samples inside manifold; 3) the discriminator objective does not encourage realization of condition (3) in Assumption 1 as discussed in Section 3.2. Our approach aims to explicitly address the above drawbacks.

Following prior work [16, 6], we employ a GAN-like implicit generator. We first sample a latent variable $z$ from a uniform distribution $\mathcal{U}(0, 1)$ for each dimension, and then apply a deep convolutional network to transform $z$ to a sample $x$.

### 5.1 Generator Entropy

Fundamentally, the first drawback concerns the entropy of the distribution of generated features, $\mathcal{H}(p_G(f))$. This connection is rather intuitive, as the collapsing issue is a clear sign of low entropy. Therefore, to avoid collapsing and increase coverage, we consider explicitly increasing the entropy.

Although the idea sounds simple and straightforward, there are two practical challenges. Firstly, as implicit generative models, GANs only provide samples rather than an analytic density form. As a result, we cannot evaluate the entropy exactly, which rules out the possibility of naive optimization. More problematically, the entropy is defined in a high-dimensional feature space, which is changing dynamically throughout the training process. Consequently, it is difficult to estimate and optimize the generator entropy in the feature space in a stable and reliable way. Faced with these difficulties, we consider two practical solutions.

The first method is inspired by the fact that input space is essentially static, where estimating and optimizing the counterpart quantities would be much more feasible. Hence, we instead increase the generator entropy in the *input space*, i.e., $\mathcal{H}(p_G(x))$, using a technique derived from an information theoretical perspective and relies on variational inference (VI). Specially, let $\mathcal{Z}$ be the latent variable space, and $\mathcal{X}$ be the input space. We introduce an additional encoder, $q : \mathcal{X} \mapsto \mathcal{Z}$, to define a variational upper bound of the negative entropy [3], $-\mathcal{H}(p_G(x)) \leq -\mathbb{E}_{x,z \sim p_G} \log q(z|x) = L_{\text{VI}}$. Hence, minimizing the upper bound $L_{\text{VI}}$ effectively increases the generator entropy. In our implementation, we formulate $q$ as a diagonal Gaussian with bounded variance, i.e. $q(z|x) = \mathcal{N}(\mu(x), \sigma^2(x))$, with $0 < \sigma(x) < \theta$, where $\mu(\cdot)$ and $\sigma(\cdot)$ are neural networks, and $\theta$ is the threshold to prevent arbitrarily large variance.

Alternatively, the second method aims at increasing the generator entropy in the feature space by optimizing an auxiliary objective. Concretely, we adapt the pull-away term (PT) [25] as the auxiliary cost, $L_{\text{PT}} = \frac{1}{N(N-1)} \sum_{i=1}^{N} \sum_{j \neq i} \left( \frac{f(x_i)^\top f(x_j)}{\|f(x_i)\|\|f(x_j)\|} \right)^2$, where $N$ is the size of a mini-batch and $x$ are samples. Intuitively, the pull-away term tries to orthogonalize the features in each mini-batch by minimizing the squared cosine similarity. Hence, it has the effect of increasing the diversity of generated features and thus the generator entropy.

### 5.2 Generating Low-Density Samples

The second drawback of feature matching GANs is that high-density samples can be generated in the feature space, which is not desirable according to our analysis. Similar to the argument in Section 5.1, it is infeasible to directly minimize the density of generated features. Instead, we enforce the generation of samples with low density in the input space. Specifically, given a threshold $\epsilon$, we minimize the following term as part of our objective:

$$\mathbb{E}_{x \sim p_G} \log p(x) \mathbb{I}[p(x) > \epsilon] \tag{3}$$

where $\mathbb{I}[\cdot]$ is an indicator function. Using a threshold $\epsilon$, we ensure that only high-density samples are penalized while low-density samples are unaffected. Intuitively, this objective pushes the generated samples to "move" towards low-density regions defined by $p(x)$. To model the probability distribution over images, we simply adapt the state-of-the-art density estimation model for natural images, namely the PixelCNN++ [17] model. The PixelCNN++ model is used to estimate the density $p(x)$ in Eq. (3). The model is pretrained on the training set, and fixed during semi-supervised training.

## 5.3 Generator Objective and Interpretation

Combining our solutions to the first two drawbacks of feature matching GANs, we have the following objective function of the generator:

$$\min_G \quad -\mathcal{H}(p_G) + \mathbb{E}_{x \sim p_G} \log p(x) \mathbb{I}[p(x) > \epsilon] + \|\mathbb{E}_{x \sim p_G} f(x) - \mathbb{E}_{x \sim \mathcal{U}} f(x)\|^2. \tag{4}$$

This objective is closely related to the idea of complement generator discussed in Section 3. To see that, let's first define a target complement distribution in the input space as follows

$$p^*(x) = \begin{cases} \frac{1}{Z} \frac{1}{p(x)} & \text{if } p(x) > \epsilon \text{ and } x \in \mathcal{B}_x \\ C & \text{if } p(x) \leq \epsilon \text{ and } x \in \mathcal{B}_x, \end{cases}$$

where $Z$ is a normalizer, $C$ is a constant, and $\mathcal{B}_x$ is the set defined by mapping $\mathcal{B}$ from the feature space to the input space. With the definition, the KL divergence (KLD) between $p_G(x)$ and $p^*(x)$ is

$$\text{KL}(p_G \| p^*) = -\mathcal{H}(p_G) + \mathbb{E}_{x \sim p_G} \log p(x) \mathbb{I}[p(x) > \epsilon] + \mathbb{E}_{x \sim p_G} \big( \mathbb{I}[p(x) > \epsilon] \log Z - \mathbb{I}[p(x) \leq \epsilon] \log C \big).$$

The form of the KLD immediately reveals the aforementioned connection. Firstly, the KLD shares two exactly the same terms with the generator objective (4). Secondly, while $p^*(x)$ is only defined in $\mathcal{B}_x$, there is not such a hard constraint on $p_G(x)$. However, the feature matching term in Eq. (4) can be seen as softly enforcing this constraint by bringing generated samples "close" to the true data (Cf. Section 4). Moreover, because the identity function $\mathbb{I}[\cdot]$ has zero gradient almost everywhere, the last term in KLD would not contribute any informative gradient to the generator. In summary, optimizing our proposed objective (4) can be understood as minimizing the KL divergence between the generator distribution and a desired complement distribution, which connects our practical solution to our theoretical analysis.

## 5.4 Conditional Entropy

In order for the complement generator to work, according to condition (3) in Assumption 1, the discriminator needs to have strong true-fake belief on unlabeled data, i.e., $\max_{k=1}^{K} w_k^\top f(x) > 0$. However, the objective function of the discriminator in [16] does not enforce a dominant class. Instead, it only needs $\sum_{k=1}^{K} P_D(k|x) > P_D(K+1|x)$ to obtain a correct decision boundary, while the probabilities $P_D(k|x)$ for $k \leq K$ can possibly be uniformly distributed. To guarantee the strong true-fake belief in the optimal conditions, we add a conditional entropy term to the discriminator objective and it becomes,

$$\max_D \quad \mathbb{E}_{x,y \sim \mathcal{L}} \log p_D(y|x, y \leq K) + \mathbb{E}_{x \sim \mathcal{U}} \log p_D(y \leq K|x) +$$
$$\mathbb{E}_{x \sim p_G} \log p_D(K+1|x) + \mathbb{E}_{x \sim \mathcal{U}} \sum_{k=1}^{K} p_D(k|x) \log p_D(k|x). \tag{5}$$

By optimizing Eq. (5), the discriminator is encouraged to satisfy condition (3) in Assumption 1. Note that the same conditional entropy term has been used in other semi-supervised learning methods [19, 13] as well, but here we motivate the minimization of conditional entropy based on our theoretical analysis of GAN-based semi-supervised learning.

To train the networks, we alternatively update the generator and the discriminator to optimize Eq. (4) and Eq. (5) based on mini-batches. If an encoder is used to maximize $\mathcal{H}(p_G)$, the encoder and the generator are updated at the same time.

# 6 Experiments

We mainly consider three widely used benchmark datasets, namely MNIST, SVHN, and CIFAR-10. As in previous work, we randomly sample 100, 1,000, and 4,000 labeled samples for MNIST, SVHN,

| Methods | MNIST (# errors) | SVHN (% errors) | CIFAR-10 (% errors) |
|---|---|---|---|
| CatGAN [19] | $191 \pm 10$ | - | $19.58 \pm 0.46$ |
| SDGM [12] | $132 \pm 7$ | $16.61 \pm 0.24$ | - |
| Ladder network [15] | $106 \pm 37$ | - | $20.40 \pm 0.47$ |
| ADGM [12] | $96 \pm 2$ | 22.86 | - |
| FM [16] * | $93 \pm 6.5$ | $8.11 \pm 1.3$ | $18.63 \pm 2.32$ |
| ALI [4] | - | $7.42 \pm 0.65$ | $17.99 \pm 1.62$ |
| VAT small [13] * | 136 | 6.83 | 14.87 |
| Our best model * | $\mathbf{79.5 \pm 9.8}$ | $\mathbf{4.25 \pm 0.03}$ | $\mathbf{14.41 \pm 0.30}$ |
| Triple GAN [11] *‡ | $91 \pm 58$ | $5.77 \pm 0.17$ | $16.99 \pm 0.36$ |
| Π model [9] †‡ | - | $5.43 \pm 0.25$ | $16.55 \pm 0.29$ |
| VAT+EntMin+Large [13]† | - | 4.28 | 13.15 |

Table 1: Comparison with state-of-the-art methods on three benchmark datasets. Only methods without data augmentation are included. ∗ indicates using the same (small) discriminator architecture, † indicates using a larger discriminator architecture, and ‡ means self-ensembling.

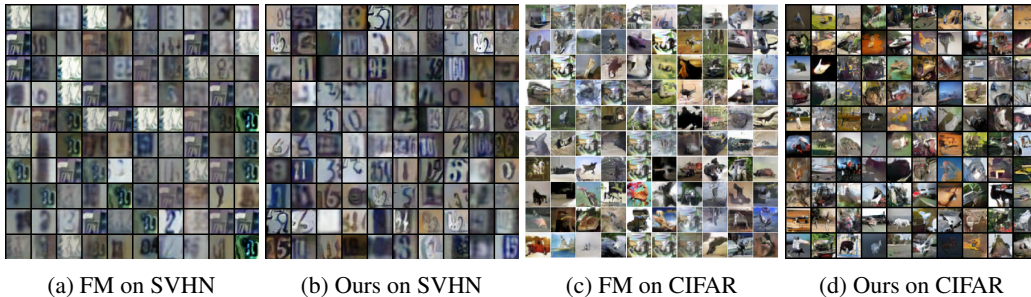

(a) FM on SVHN     (b) Ours on SVHN     (c) FM on CIFAR     (d) Ours on CIFAR

Figure 5: Comparing images generated by FM and our model. FM generates collapsed samples, while our model generates diverse "bad" samples.

and CIFAR-10 respectively during training, and use the standard data split for testing. We use the 10-quantile log probability to define the threshold $\epsilon$ in Eq. (4). We add instance noise to the input of the discriminator [1, 18], and use spatial dropout [20] to obtain faster convergence. Except for these two modifications, we use the same neural network architecture as in [16]. For fair comparison, we also report the performance of our FM implementation with the aforementioned differences.

## 6.1 Main Results

We compare the the results of our best model with state-of-the-art methods on the benchmarks in Table 1. Our proposed methods consistently improve the performance upon feature matching. We achieve new state-of-the-art results on all the datasets when only small discriminator architecture is considered. Our results are also state-of-the-art on MNIST and SVHN among all single-model results, even when compared with methods using self-ensembling and large discriminator architectures. Finally, note that because our method is actually orthogonal to VAT [13], combining VAT with our presented approach should yield further performance improvement in practice.

## 6.2 Ablation Study

We report the results of ablation study in Table 2. In the following, we analyze the effects of several components in our model, subject to the intrinsic features of different datasets.

First, the generator entropy terms (VI and PT) (Section 5.1) improve the performance on SVHN and CIFAR by up to 2.2 points in terms of error rate. Moreover, as shown in Fig 5, our model significantly reduces the collapsing effects present in the samples generated by FM, which also indicates that maximizing the generator entropy is beneficial. On MNIST, probably due to its simplicity, no collapsing phenomenon was observed with vanilla FM training [16] or in our setting. Under such circumstances, maximizing the generator entropy seems to be unnecessary, and the estimation bias introduced by approximation techniques can even hurt the performance.

| Setting | Error | Setting | Error |
|---|---|---|---|
| MNIST FM | $85.0 \pm 11.7$ | CIFAR FM | 16.14 |
| MNIST FM+VI | $86.5 \pm 10.6$ | CIFAR FM+VI | 14.41 |
| MNIST FM+LD | $79.5 \pm 9.8$ | CIFAR FM+VI+Ent | 15.82 |
| MNIST FM+LD+Ent | $89.2 \pm 10.5$ | | |

| Setting | Error | Setting | Max log-p |
|---|---|---|---|
| SVHN FM | 6.83 | MNIST FM | -297 |
| SVHN FM+VI | 5.29 | MNIST FM+LD | -659 |
| SVHN FM+PT | 4.63 | SVHN FM+PT+Ent | -5809 |
| SVHN FM+PT+Ent | 4.25 | SVHN FM+PT+LD+Ent | -5919 |
| SVHN FM+PT+LD+Ent | 4.19 | SVHN 10-quant | -5622 |

| Setting $\epsilon$ as $q$-th centile | $q = 2$ | $q = 10$ | $q = 20$ | $q = 100$ |
|---|---|---|---|---|
| Error on MNIST | $77.7 \pm 6.1$ | $79.5 \pm 9.8$ | $80.1 \pm 9.6$ | $85.0 \pm 11.7$ |

Table 2: Ablation study. *FM* is feature matching. *LD* is the low-density enforcement term in Eq. (3). *VI* and *PT* are two entropy maximization methods described in Section 5.1. *Ent* means the conditional entropy term in Eq. (5). *Max log-p* is the maximum log probability of generated samples, evaluated by a PixelCNN++ model. *10-quant* shows the 10-quantile of true image log probability. *Error* means the number of misclassified examples on MNIST, and error rate (%) on others.

Second, the low-density (LD) term is useful when FM indeed generates samples in high-density areas. MNIST is a typical example in this case. When trained with FM, most of the generated hand written digits are highly realistic and have high log probabilities according to the density model (Cf. max log-p in Table 2). Hence, when applied to MNIST, LD improves the performance by a clear margin. By contrast, few of the generated SVHN images are realistic (Cf. Fig. 5a). Quantitatively, SVHN samples are assigned very low log probabilities (Cf. Table 2). As expected, LD has a negligible effect on the performance for SVHN. Moreover, the "max log-p" column in Table 2 shows that while LD can reduce the maximum log probability of the generated MNIST samples by a large margin, it does not yield noticeable difference on SVHN. This further justifies our analysis. Based on the above conclusion, we conjecture LD would not help on CIFAR where sample quality is even lower. Thus, we did not train a density model on CIFAR due to the limit of computational resources.

Third, adding the conditional entropy term has mixed effects on different datasets. While the conditional entropy (Ent) is an important factor of achieving the best performance on SVHN, it hurts the performance on MNIST and CIFAR. One possible explanation relates to the classic exploitation-exploration tradeoff, where minimizing Ent favors exploitation and minimizing the classification loss favors exploration. During the initial phase of training, the discriminator is relatively *uncertain* and thus the gradient of the Ent term might dominate. As a result, the discriminator learns to be more confident even on incorrect predictions, and thus gets trapped in local minima.

Lastly, we vary the values of the hyper-parameter $\epsilon$ in Eq. (4). As shown at the bottom of Table 2, reducing $\epsilon$ clearly leads to better performance, which further justifies our analysis in Sections 4 and 3 that off-manifold samples are favorable.

## 6.3 Generated Samples

We compare the generated samples of FM and our approach in Fig. 5. The FM images in Fig. 5c are extracted from previous work [16]. While collapsing is widely observed in FM samples, our model generates diverse "bad" images, which is consistent with our analysis.

## 7 Conclusions

In this work, we present a semi-supervised learning framework that uses generated data to boost task performance. Under this framework, we characterize the properties of various generators and theoretically prove that a complementary (i.e. bad) generator improves generalization. Empirically our proposed method improves the performance of image classification on several benchmark datasets.

## Acknowledgement

This work was supported by the DARPA award D17AP00001, the Google focused award, and the Nvidia NVAIL award. The authors would also like to thank Han Zhao for his insightful feedback.

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
