[Supplementary Material · camera_ready_appendix.pdf]

# 8   Appendix

## 8.1   Proof of Proposition 1

*Proof.* Given an optimal solution $D = (w, f)$ for the supervised objective, due to the infinite capacity of the discriminator, there exists $D^* = (w^*, f^*)$ such that for all $x$ and $k \leq K$,

$$\exp(w_k^{*\top} f^*(x)) = \frac{\exp(w_k^\top f(x))}{\sum_{k'} \exp(w_{k'}^\top f(x))} \tag{6}$$

For all $x$,

$$P_{D^*}(y|x, y \leq K) = \frac{\exp(w_k^{*\top} f^*(x))}{\sum_{k'} \exp(w_{k'}^{*\top} f^*(x))} = \frac{\exp(w_k^\top f(x))}{\sum_{k'} \exp(w_{k'}^\top f(x))} = P_D(y|x, y \leq K)$$

Let $L_D$ be the supervised objective in Eq. (1). Since $p = p_G$, the objective in Eq. (1) can be written as

$$J_D = L_D + \mathbb{E}_{x \sim p} \left[ \log P_D(K+1|x) + \log(1 - P_D(K+1|x)) \right]$$

Given Eq. (6), we have

$$P_{D^*}(K+1|x) = \frac{1}{1 + \sum_k \exp w_k^{*\top} f^*(x)} = \frac{1}{2}$$

Therefore, $D^*$ maximizes the second term of $J_D$. Because $D$ maximizes $L_D$, $D^*$ also maximizes $L_D$. It follows that $D^*$ maximizes $J_D$. □

## 8.2   On the Feature Space Bound Assumption

To obtain our theoretical results, we assume that $\cup_{k=1}^K F_k$ is bounded by a convex set $\mathcal{B}$. And the definition of complement generator requires that $F_G = \mathcal{B} - \cup_{k=1}^K F_k$. Now we justify the necessity of the introduction of $\mathcal{B}$.

The bounded $\mathcal{B}$ is introduced to ensure that Assumption 1 is realizable. We first show that for Assumption 1 to hold, $F_G$ must be a convex set.

We define $S = \{f : \max_{k=1}^K w_k^\top f < 0\}$.

**Lemma 2.** *$S$ is a convex set.*

*Proof.* We prove it by contradiction. Suppose $S$ is a non-convex set, then there exists $f_1, f_2 \in S$, and $0 < \alpha < 1$, such that $f = \alpha f_1 + (1 - \alpha) f_2 \notin S$. For all $k$, we have $w_k^\top f_1 < 0$ and $w_k^\top f_2 < 0$, and thus it follows

$$w_k^\top f = \alpha w_k^\top f_1 + (1 - \alpha) w_k^\top f_2 < 0$$

Therefore, $\max_{k=1}^K w_k^\top f < 0$, and we have $f \in S$, leading to contradiction.

We conclude that $S$ is a convex set. □

If the feature space is unbounded and $F_G$ is defined as $\mathbb{R}^d - \cup_{k=1}^K F_k$, where $d$ is the feature space dimension, then by Assumption 1, we have $S = F_G$. Since $F_G$ is the complement set of $\cup_{k=1}^K F_k$ and $F_k$'s are disjoint, $F_G$ is a non-convex set, if $K \geq 2$. However, by Lemma 2, $F_G$ is convex, leading to contradiction. We therefore define the complement generator using a bound $\mathcal{B}$.

## 8.3   The Reasonableness of Assumption 1

Here, we justify the proposed Assumption 1.

**Classification correctness on $\mathcal{L}$**   For (1), it assumes the correctness of classification on labeled data $\mathcal{L}$. This only requires the transformation $f(x)$ to have high enough capacity, such that the *limited amount* of labeled data points are linearly separable in the feature space. Under the setting of semi-supervised learning, where $|\mathcal{L}|$ is quite limited, this assumption is usually reasonable.

**True-Fake correctness on $\mathcal{G}$**   For (2), it assumes that on generated data, the classifier can correctly distinguish between true and generated data. This can be seen by noticing that $w_{K+1}^\top f = 0$, and the assumption thus reduces to $w_{K+1}^\top f(x) > \max_{k=1}^K w_k^\top f(x)$. For this part to hold, again we essentially require a transformation $f(x)$ with high enough capacity to distinguish true and fake data, which is a standard assumption made in GAN literature.

**Strong true-fake belief on $\mathcal{U}$**   Finally, part (3) of the assumption is a little bit trickier than the other two.

- Firstly, note that (3) is related to the true-fake correctness, because $\max_{k=1}^K w_k^\top f(x) > 0 = w_{K+1}^\top f(x)$ is a *sufficient* (but not necessary) condition for $x$ being classified as a true data point. Instead, the actual necessary condition is that $\log \sum_{k=1}^K \exp(w_k^\top f(x)) \geq w_{K+1}^\top f(x) = 0$. Thus, it means the condition (3) might be violated.

- However, using the relationship $\log \sum_{k=1}^K \exp(w_k^\top f(x)) \leq \log K \max_{k=1}^K \exp(w_k^\top f(x))$, to guarantee the necessary condition $\log \sum_{k=1}^K \exp(w_k^\top f(x)) \geq 0$, we must have

$$\log K \max_{k=1}^K \exp(w_k^\top f(x)) \geq 0$$

$$\implies \max_{k=1}^K w_k^\top f(x) \geq \log 1/K$$

  Hence, if the condition (3) is violated, it means

$$\log 1/K \leq \max_{k=1}^K w_k^\top f(x) \leq 0$$

  Note that this is a very small interval for the logit $w_k^\top f(x)$, whose possible range expands the entire real line $(-\infty, \infty)$. Thus, the region where such violation happens should be limited in size, making the assumption reasonable in practice.

- Moreover, even there exists a limited violation region, as long as part (1) and part (2) in Assumption 1 hold, Proposition 2 always hold for regions inside $\mathcal{U}$ where $\max_{k=1}^K w_k^\top f(x) > 0$. This can be viewed as a further Corollary.

Figure 6: Percentage of the test samples that satisfy the assumption under our best model.

Empirically, we find that it is easy for the model to satisfy the correctness assumption on labeled data perfectly. To verify the other two assumptions, we keep track of the percentage of test samples that the two assumptions hold under our best models. More specifically, to verify the true-fake correctness on $\mathcal{G}$, we calculate the ratio after each epoch

$$\frac{\sum_{x \sim \mathcal{T}} \mathbb{I}[\max_{i=1}^K w_i^\top f(x) > 0]}{|\mathcal{T}|},$$

where $\mathcal{T}$ denotes the test set and $|\mathcal{T}|$ is number of sample in it. Similarly, for the strong true-fake belief on $\mathcal{U}$, we generate the same number of samples as $|\mathcal{T}|$ and calculate

$$\frac{\sum_{x \sim p_G} \mathbb{I}[\max_i w_i^T f(x) < 0]}{|\mathcal{T}|}$$

The plot is presented in Fig. 6. As we can see, the two ratios are both above $0.9$ for both SVHN and CIFAR-10, which suggests our assumptions are reasonable in practice.

### 8.4 Proof of Lemma 1

*Proof.* Let $\Delta f = f_G - f'_G$, then we have $\|\Delta f\|_2 \leq \epsilon$. Because $w_k^\top f'_G < 0$ by assumption, it follows

$$w_k^\top f_G = w_k^\top (f'_G + \Delta f) = w_k^\top f'_G + w_k^\top \Delta f < w_k^\top \Delta f \leq C\epsilon$$

$\square$