[Reviews · NeurIPS 2017]

Reviewer 1



This work extends and improves the performance of GAN based approaches to semi-supervised learning as explored in both "Improved techniques for training gans" (Salimans 2016) and "Unsupervised and semi-supervised learning with categorical generative adversarial networks" (Springenberg 2015). The paper introduces the notion of a complement generator which tries to sample from low-density areas of the data distribution (in feature space) and explores a variety of objective terms motivated/connected to this analysis. It is difficult to exactly match the motivated objective, due to various hard to estimate quantities like density and entropy, the paper uses a variety of approximations in their place. In addition to an illustrative case study on synthetic data, the paper has a suite of experiments on standardized semi-supervised tests including ablation studies on the various terms proposed. The overall empirical results are a significant improvement over the Feature Matching criteria proposed in "Improved techniques for training gans". Given the variety of objective terms suggested it is important and appreciated that the authors included ablation studies. It is unfortunate that they are not completely thorough, however. For instance, why does SVHN have a 5 different experiments but MNIST 2 and CIFAR-10 3? Why are the Approximate Entropy Maximization terms only tested on SVHN? How do they perform on CIFAR-10? Could the author(s) comment on why a fuller suite of comparisons was not completed? The benefits of the various terms are not always consistent across datasets. The text mentions and discusses this briefly but a more thorough investigation by completing the ablation studies would be helpful for people wishing to extend/improve upon the ideas presented in this paper.

Reviewer 2



In this paper, the authors proposed a novel semi-supervised learning algorithm based on GANs. It demonstrated that given the discriminator objective, good semi-supervised learning indeed requires a bad generator. They derive a novel formulation that substantially improves over feature matching GANs. Experiments demonstrate the state-of-the-art results on multiple benchmark datasets. The paper is interesting and well-written, which is important for the family of GANs algorithm. I have several concerns as follows: 1. As is demonstrate in W-GAN, the GANs is not easy to converge when we minimizing the KL divergence between the generator distribution and the target distribution. It would be nice if the authors could demonstrate whether the algorithms can be applied to more advanced GANs. 2. As is well-known, it is not easy to generate the general samples based on GANs, such as the images with high resolutions. It is expected the author could demonstrate how to use the proposed models when sample generation is not easy. 3. It would be nice if the authors could demonstrate how “bad” a generator should be for a semi-supervised classifier. 4. It is interesting if the authors could demonstrate whether the human-specified labels can improve the performance of the generator, such that some satisfactory images can be generated.

Reviewer 3



After reading the rebuttal I changed my score to 7. Overall it is an interesting paper with an interesting idea. Although the theoretical contributions are emphasized I find the empirical findings more appealing. The theory presented in the paper is not convincing (input versus feature, convexity etc). I think the link to classical semi-supervised learning and the cluster assumption should be emphasized, and the * low density assumption on the boundary* as explained in this paper : Semi-Supervised Classification by Low Density Separation Olivier Chapelle, Alexander Zien http://citeseerx.ist.psu.edu/viewdoc/download?doi=10.1.1.76.5826&rep=rep1&type=pdf I am changing my review to 7, and I hope that the authors will put their contribution in the context of known work in semi-supervised learning , that the boundary of separation should lie in the low density regions . This will put the paper better in context. ----- This paper provides an analysis of how GAN helps in semi supervised learning (in the “K+1” setting of [15]). The paper outlines some assumptions under which it is shown that a complement generator is needed to improve the accuracy of the supervised task at hand. Meaning that the generator needs to target low densities areas in the input space. Using this idea an algorithm is given that combines that feature matching criterium , with a density estimation (using a pixel CNN ) under which the generator targets low density areas of the fixed estimated model (for a given threshold of log proba,). Other entropy regularizers are added to encourage diversity in the generator. Positive empirical results are reported. Understanding GANs in the semi supervised setting and improving it is an important problem, however the paper has many caveats: - while the analysis is done in the feature space under a lot of assumptions, the method proposed is in the input space , which gives a big mismatch between the analysis and the proposed method . Convexity that is used all over the proofs is not any more valid. - the KL expression (line 238 ) is wrong: the term assumed to be constant is not constant. It is equal to ‘-log(C) P(x sim p_g, p(x)<=epsilon)’, this term should be optimized as well. Some other way to come up at the objective presented need to be developed. Maybe just motivating the minimization of the cross entropy term, and adding a cross entropy regularization? - From the analysis and the 2D example with uniform sampling off the manifold, the paper seems to suggest that the generator should supply samples outside the manifold, in a sense it reinforces the boundaries of the classifier by providing only negative samples. The truth is in between: the generator should not provides too strong samples (very realistic, easy to classify as a class ) , nor too weak samples that are easily labeled as fake. it should be portion on the manifolds to reinforce the positive , and portions outside to reinforce the negative. A more realistic setup may be probabilistic where the assumption are assumed to hold with probability 1- delta as off the manifold, and delta on the manifold, although corollary 1 is not proven and hard to understand how it could hold , it seems not realistic to me . Balancing this delta seem to be crucial and seems inline with the analysis of https://arxiv.org/pdf/1705.08850.pdf - Ablation study: Performance of the method in section 6 for many values of epsilon values would illustrate the discussion above, and should be reported, also a justification of the 10 quantile should be provided. -Overall using a density estimation (Pixel CNN) and entropic regularizer with another estimator seems a bit adhoc and not satisfying, wonder if the authors have other thoughts to avoid those estimators while encouraging a form of 'low density sampling’ while using only implicit modeling. or hybrid implicit/ prescribed modeling while maintaining an end to end training? Minor comments: - the loss is L_supervised + L_unsupervised have you experimented with balancing the two terms with a penalty term lambda, how does this balancing interact with the threshold epsilon? - In lines '162 and 163' This builds a connection with ... [2,25] which leverage "the low density boundary assumption". I don't understand this comment , can you elaborate more on what this assumption in laplacian regularization for instance? - line 178 typo : 'the a bounded' - corollary one : the proof is not provided. I don't understand how those claims are true uniformly over all X, and what does it means |G| goes to infinity, you may need stronger assumptions here....